applied mathematics/mathematical modelling/ differential equations

running, track, optimal control, neural drive, energy, force

**Author for correspondence:**
Amandine Aftalion
e-mail: amandine.aftalion@ehess.fr

# How to build a new athletic track to break records

Amandine Aftalion[1] and Emmanuel Trélat[2]

[1]Ecole des Hautes Etudes en Sciences Sociales, Centre d'Analyse et de Mathématique Sociales UMR-8557, Paris, France
[2]Sorbonne Université, CNRS, Université de Paris, Inria, Laboratoire Jacques-Louis Lions (LJLL), 75005 Paris, France

 AA, 0000-0002-1936-3798

We introduce a new optimal control model which encompasses pace optimization and motor control effort for a runner on a fixed distance. The system couples mechanics, energetics, neural drive to an economic decision theory of cost and benefit. We find how effort is minimized to produce the best running strategy, in particular, in the bend. This allows us to discriminate between different types of tracks and estimate the discrepancy between lanes. Relating this model to the optimal path problem called the Dubins path, we are able to determine the geometry of the optimal track and estimate record times.

## 1. Introduction

Usain Bolt's 200 m record has not been beaten for 10 years and Florence Griffith Joyner's for more than 30 years. What about if the secret behind beating records was to build a new athletic track with a better geometry? Researchers have addressed theoretical issues on various aspects of sport records [1–5] or strategies [6–10], on the effect of running on a bend [11–15], however, little has been done on how to improve the records for running 200 m. Indeed, the more economical way to run is on a straight, but only the 100 m is run straight. Starting from 200 m, the track has curved parts so that the runner has to counter the centrifugal force and inner lanes are therefore disadvantaged. Analysing what is the geometry of an optimal track to beat records has never, to our knowledge, been addressed. Here, we introduce an optimal control problem (OCP) to build a track related to the well-known Dubins problem [16]. Introducing a new model for pace optimization and motor control effort, we determine the optimal running strategy which leads to the design of a new track having shorter straights and larger radii.

It is not that it is impossible to build a straight 200 m or 400 m track, but it would not be convenient for the audience and it would make a poor arena. At present, there are three designs of tracks which can be certified by the International Association of Athletics Federations (IAAF) [17]: standard tracks (made of straights and semi-circles) and two types of double bend tracks (where the double bend is made of three arcs of two different radii) as

**Figure 1.** Geometry and dimensions for IAAF certified tracks. The straight length is $AB$, and the width is $BC$. The standard track is made of two straights and two semi-circles. There are two types of double bend tracks (DB1 and DB2) and each of them has a smaller outside radius than the standard track, leading to a higher centrifugal force. Note that the distance 400 m is achieved at 0.3 m from the boundary of the first lane.

illustrated in figure 1. It is usually admitted in the athletic community that the standard track is the quickest and there is no hope to beat a record on a double bend track, the second type double bend 2 (DB2) with the longest straight being the worst. Actually, the double bend tracks have been designed to include a football or rugby stadium in it, and the big disadvantage is that the bends have a smaller radius of curvature. Therefore, the centrifugal force is bigger and the double bend tracks are slower in total [11,18]. Moreover, on such tracks, there is a major disadvantage on being on lane 1, or the inside lanes where the curvature is the largest, because there is a bigger difference between extreme lanes than on standard tracks. So the multi-sports arenas are certainly not favourable for athletic records! The issue of this paper is to analyse better what is an optimal track to beat records, and also a track which minimizes the disadvantage of inner lanes.

When on a bend, the runner uses his propulsive force both to move along the track and to act against the centrifugal force $mv^2/R$, where $m$ is his mass, $v$ is his velocity and $R$ is the radius of curvature of the track (which depends on the position of the runner on the track). Therefore, there is a limitation, also called constraint of movement [11] which is

$$f^2 + \frac{v^4}{R^2} \le f_{\max}^2,$$

where $f_{\max}$ is the maximal propulsive force per unit mass and $f$ is the time-dependent propulsive force per unit mass in the direction of movement. A first idea is to find an optimal track without solving in detail the runner's equations of movement but minimizing the maximum of the curvature (that is $1/R$) over the track. Indeed, this way, the velocity and propulsive force can be maximized. Without any extra constraint, the optimal solution minimizing the maximum of curvature is the straight [16,19], but we want to impose having a closed loop in order to have a nice arena for the audience. Another constraint can be to additionally include an inner rectangle for the multi sport activities. Denoting by $ABCD$ a track as in figure 1, we assume that the track is symmetric with respect to horizontal and vertical lines, that $(AB)$ and $(CD)$ are straights so that $\overset{\frown}{BC}$ and $\overset{\frown}{DA}$ are the bends on each side.

## 2. Optimal track

### 2.1. First problem

Assume that the length of the straight is fixed, hence the length $l_b$ of the bend is fixed too, but the Euclidean distance $BC$ between the points $B$ and $C$ is free.

Find the optimal curve of prescribed length $l_b$ joining two free points $B$ and $C$, with horizontal tangent at $B$ and $C$, and minimizing the upper bound of the curvature.

The problem is a particular case of the well-known *Dubins problem* [16,19] for which solutions, in general, are concatenations of straight lines and of arcs of a circle. In our case, the optimal geometry is a semi-circle leading to the standard track made of straights and semi-circles. If one ignores the obligation to have a straight of prescribed length, then the optimal arena from this point of view is a circle. Actually, it would be better to have a shorter straight than the official standard track in order to have a bigger radius for the two semi-circles and thus a smaller centrifugal force. Our simulations on the full runner problem (see below) indicate that below a length of 60 m for the straight line, the difference becomes tiny in terms of performance (of the order of one-thousandth of a second) because the centrifugal force is no longer significant. But a straight of 60 m is definitely quicker by four hundredth of a second than the present straight of 84.39 m. Therefore, the present standard track is not favourable for athletic records.

Let us be more specific about the OCP to design the best track to beat records. For fixed $l$,

$$(\text{OCP}) \begin{cases} \min \ M \text{ s.t. } |u(s)| \le M \text{ for } s \in [0, l], \\ \dot{x}(s) = \cos \theta(s), \ x(l) = 0, \\ \dot{y}(s) = \sin \theta(s), \ y(0) = 0, \\ \dot{\theta}(s) = u(s), \ \theta(0) = \frac{\pi}{2}, \ \theta(l) = \pi. \end{cases}$$

By symmetry, this represents a quarter of the bend of length $l$. The curvature of the track is given by $u(s) = \dot{\theta}(s)$ which is our control. The usual locomotion problem [20] takes $w = \dot{u}$ as a control and minimizes $\int_0^l w(s)^2 \, ds$, but here we want to minimize the maximum of the curvature. It follows from [16,19] that the solution is unique and is a quarter of a circle. Without any additional constraint, this leads to the standard track.

## 2.2. Second problem

Because of the constraint to include a football or a rugby stadium inside the track, the Euclidean distance $BC$ has to be greater than that realizing the minimum of problem 1. In other words, the value $y(l)$ is prescribed. Then of course, the semi-circle is not possible because the length of the curve and the distance $BC$ are not consistent.

Find the optimal curve of prescribed length $l_b$ joining two free points $B$ and $C$ whose distance is bigger than some prescribed value $l_2$, with horizontal tangent at $B$ and $C$, minimizing the upper bound of the curvature, and so that the track $ABCD$ contains a fixed stadium.

It follows from [16] that the optimal solution is unique and made up of two-quarters of a circle joined by a vertical straight, instead of three arcs of a circle as in the double bend tracks. If $l_b$ is the length of the bend given by figure 1, $R$ is the radius and $l_v$ is the length of the vertical straight, then

$$2R + l_v = BC \quad \text{and} \quad \pi(R + 0.3) + l_v = l_b.$$

Note that we have to add 0.3 m to compute the length because the runner is supposed to run the whole track of 400 m at 0.3 m of the border. Therefore, $(\pi - 2)R = l_b - BC - 0.3\pi$. For double bend 1 (DB1), we fix $AB = 79.996$ and want to include a football stadium of $70 \times 109$ so that $BC \ge 70$. The optimization yields $R = 34.22$ (which is very close to the outer radius of DB1 which is 34) and $l_v = 11.56$. For DB2, we fix $AB = 98.52$ and $BC \ge 72$, then the optimization yields $R = 25$ (slightly bigger than the outer radius for DB2 which is 24) and $l_v = 22$. In figure 2, we see that problem 2 yields for the constraints of DB2 a solution which is very close to the actual design of the track.

## 2.3. Third problem

Take a fixed rectangle or a fixed shape corresponding to a football stadium or a rugby stadium. Find the optimal curve of length 400 m, encompassing this shape, at a distance at least 0.3 m, and minimizing the maximum of the curvature.

The difference with problem 2 is that we do not impose a straight of fixed length.

The football stadium is a rectangle of length 109 m and width 70 m. The rugby stadium has to include a rectangle of length 95 m and width 73 m, to which are added two small rectangles on each side of length 12 m and width 66 m, leading to a new rectangle of size $119 \times 66$.

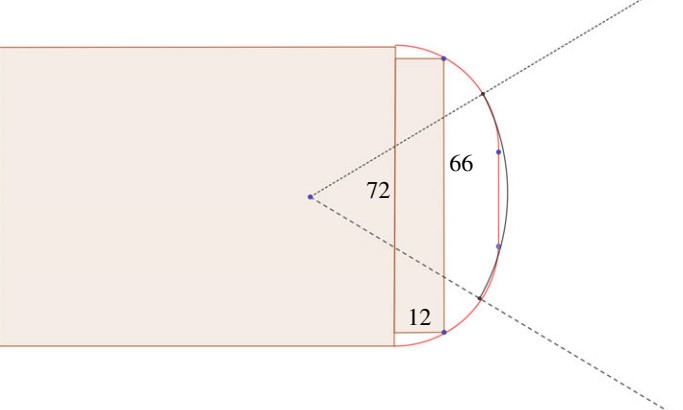

**Figure 2.** Optimal track (red) for a fixed distance $AB = 98.52$ and for $BC \geq 72$, including a rugby stadium which are the grey rectangles. There are two-quarters of a circle, separated by a vertical straight, which is eventually very close to the double bend track in black.

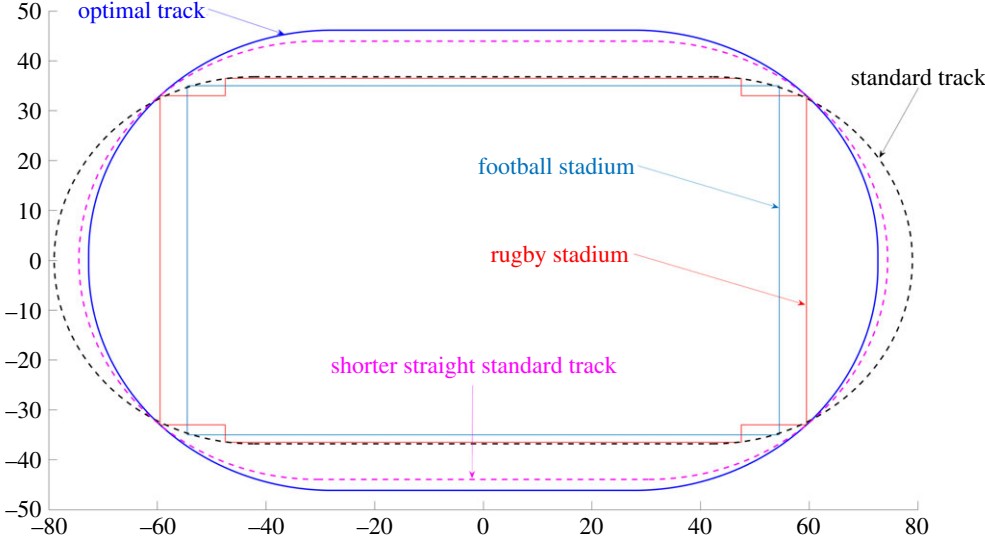

**Figure 3.** Optimal track (blue) including a rugby stadium. It is made of horizontal lines of length 55.52 m, of vertical lines of length 2.52 m and of quarters of a circle of radius 44.53 m. It is wider by 8 m than the standard track (dotted black), which contains neither the football stadium nor the rugby stadium. The optimal track will allow the breaking of records. Also plotted is a standard track with shorter straights (dotted magenta) which also includes a football and a rugby stadium; it consists of two 61 m straights and two semi-circles.

An extra constraint is then to have a track encompassing a fixed rectangle. For instance, define a rectangle of width $2\,l_1$ and length $2\,l_2$, and optimize the curve, as above, under the additional constraint

$$\max\left(\frac{|x(s)|}{l_1}, \frac{|y(s)|}{l_2}\right) \geq 1 \quad \forall s \in [0, l].$$

Numerically, with the above parameter values, we observe that such a track is unique. It is made up of straight lines joined by quarters of a circle. The contact point with the inner rectangle is on the quarter circle so that the length is computed as

$$\pi(R + 0.3) + l_v + l_h = 200,$$

where $l_v$, $l_h$ are, respectively, the lengths of the vertical and horizontal lines. This yields the track solving problem 3 illustrated in figure 3.

The optimal curve, solution to problem 3, is a wider track than the usual tracks, as illustrated in figure 3. This optimal track is much better for beating records as we will see below.

# 3. Runner model

In order to check that the optimal track solving problem 3 illustrated in figure 3 or a standard track with shorter straights are indeed significantly better than the existing ones, we next estimate the runner's performance on these tracks. Our aim is to prove that, on such an optimized track, not only good runners can improve their records, but also, the discrepancy between extreme lanes is decreased. Our numerical simulations to compute the optimal time rely on solving an OCP on a curved track [11] that we improve using a neural drive model [21,22] and taking into account a restoring force to straighten back. It articulates motor control to economic decision theory of cost and benefit. We also use the minimal intervention principle [23] so that effort is minimized through penalty terms.

Let $d > 0$ be fixed, $x(t)$ be the position of the runner at time $t$, $v(t)$ the velocity, $e(t)$ the anaerobic energy, and $f(t)$ the propulsive force. The system for a runner on a straight track introduced in [24] and extended by [25–29] relies on Newton's second law and on an energy equation. To better encompass the variations of the propulsive force, we improve the previous model [11] by adding a motor control equation limiting the variations of $f(t)$ through the motor control $u(t)$ as introduced in another context [22]. The energy equation states that the power of the propulsive force is equal to the available power coming from the anaerobic energy $e(t)$ and the aerobic energy, that is the energetic equivalent of the oxygen uptake VO2 [11]. Newton's second law applied to the runner takes into account the propulsive force $f$ and the friction term, that we choose to be linear in velocity and this provides an equation in the direction of movement:

$$\dot{x}(t) = v(t) \qquad\qquad x(0) = 0, \quad x(t_f) = d$$
$$\dot{v}(t) = -\frac{v(t)}{\tau} + f(t) \qquad\qquad v(0) = v_0$$
$$\dot{f}(t) = \gamma(u(t)(F_{\max} - f(t)) - f(t)) \quad f(t) \geq 0$$
$$\dot{e}(t) = \sigma(e(t)) - f(t)v(t) \qquad e(0) = e^0, \quad e(t) \geq 0, \quad e(t_f) = 0,$$

where $e^0 > 0$ is the initial energy, $\tau > 0$ is the friction coefficient related to the runner's economy, $F_{\max} > 0$ is a threshold upper bound for the force, $\gamma > 0$ is the time constant of motor activation, and $u(t)$ is the neural drive which will be a control. The second equation is Newton's second law applied to the runner, taking into account the propulsive force $f$ and the friction term, that we choose to be linear in velocity. The third equation is the motor control equation limiting the variations of $f$ through the motor control $u$. The fourth equation is the energy equation: the power of the propulsive force is equal to the available power coming from the anaerobic energy $e(t)$ and the aerobic energy. The function $\sigma(e)$ is the energetic equivalent of the oxygen uptake VO2 [11]. For a short race, we have

$$\sigma(e) = \sigma_{\max} \frac{e^0 - e}{e^0}.$$

The OCP on a straight consists in minimizing a cost: as the runner wants to minimize his final time by optimizing his effort, the cost is the sum of the final time and a weighted $L^2$ norm of the motor control $u$. But other additional terms will be considered on the bend.

When running a bend, one has to take into account the angle $\theta(t)$ with respect to the vertical axis. Therefore, one has to project Newton's law on the Frénet vectors to find the reaction $N$ of the ground. At equilibrium, we have

$$N = g\cos\theta + \frac{v^2}{R}\sin\theta,$$

and then the angle $\theta$ is such that

$$\tan\theta = \frac{v^2}{Rg}.$$

Nevertheless, because the velocity $v$ varies with time and the radius of curvature changes abruptly at the end of the bend, we go one step further in the modelling and we consider that the runner is modelled by a rod of length $2\ell$ having an angle $\theta$ with the vertical axis. As illustrated in figure 4, in the moving frame at velocity $v$, the forces per unit mass acting on the rod are the weight $g$, the centrifugal force $f_c = v^2/R$ and the reaction of the ground $N$. Moreover, because the runner wants to anticipate the return to the straight line, he exerts a restoring force at his foot to straighten back to $\theta = 0$. This is why we take $\dot{\theta}(t)$ as another control. We need to penalize the cost with the variables we control, as well as with the angle variations.

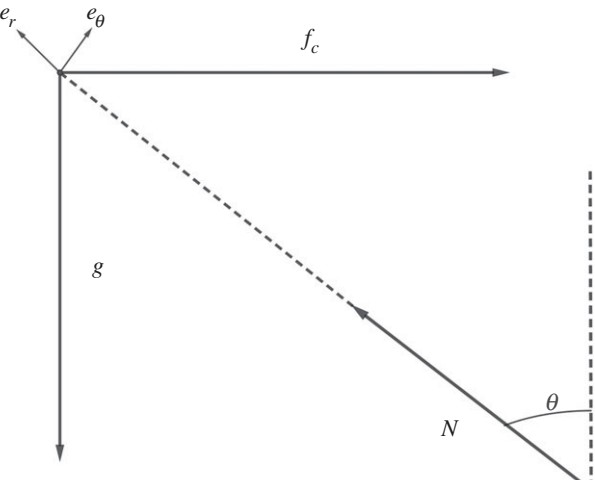

**Figure 4.** Forces per unit mass acting on a runner modelled by a rod.

This is a minimal effort principle [23]. We choose a penalty term related to the derivatives of the position in the angular plane

$$\|\ddot{r}(t)\|^2 = \ell^2\ddot{\theta}(t)^2 + \ell^2\dot{\theta}(t)^4.$$

Assuming regular variations of the control $\dot{\theta}$, we find the expression of $\ddot{\theta}$ from the sum of momentum around the axis which yields

$$\frac{J}{\ell}\ddot{\theta}(t) = \frac{v(t)^2}{R}\cos\theta(t) - g\sin\theta(t),$$

where $J = \frac{4}{3}\ell^2$ for a runner of height $2\ell$, recalling that everything is by unit of mass. Therefore, our penalty term is

$$\frac{9}{16}\left(\frac{v(t)^2}{R\cos\theta(t)} - g\sin\theta(t)\right)^2 + \ell^2\dot{\theta}(t)^4.$$

Because the runner has a finite force, we have an upper bound coupling the propulsive force, the velocity and the angle

$$f(t)^2 + N(t)^2 \le f_M^2 + g^2.$$

We find the value of $N(t)$ from the projection on $e_r$ taking into account the normal acceleration

$$\ell\dot{\theta}(t)^2 = N(t) - g\cos\theta(t) - \frac{v(t)^2}{R}\sin\theta(t).$$

We also add, in the energy equation, a term related to the power developed by the restoring force in the energy consumption, as well as a term related to the cost of staying with a static momentum. We refer to [30] for a choice of the coefficient of cost of this static moment: $c_{\text{stat}} = 0.17$.

The OCP consists in solving the solutions under the constraints by minimizing the cost: because the runner wants to minimize his final time by optimizing his effort, the cost is the sum of the final time and a penalty term owing to the controls $u$, $\dot{\theta}$ and a term related to variations of the angle. We, therefore, follow the minimal intervention principle [23] in human movement. This provides a well-posed OCP that we solve numerically.

Instead of writing the equations of motion in the time variable, we write them using the distance variable $s$. This amounts to dividing by $v$ the derivatives in time to get the derivatives in space.

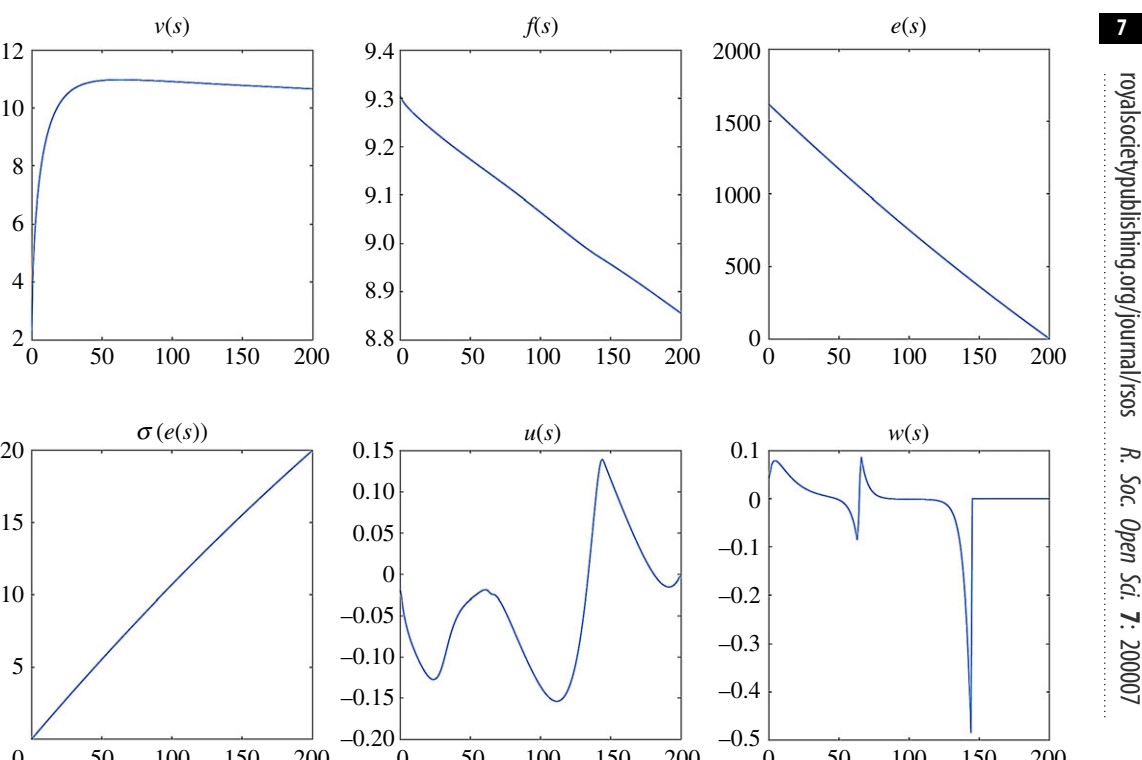

**Figure 5.** For the parameter values identified for Johnson, plot of the velocity $v(s)$ versus the distances, the propulsive force $f(s)$, the energy $e(s)$, the oxygen uptake $\sigma(e(s))$, and the controls $u(s)$ and $w(s)$.

Turning this into the distance variable, this yields the OCP

$$
\min \int_0^d \frac{1}{v(s)} \left( 1 + \varepsilon_1 u(s)^2 \right.
$$

$$
\left. + \varepsilon_2 \left( \frac{9}{16} \left( \frac{v(s)^2}{R_k(s)} \cos \theta(s) - g \sin \theta(s) \right)^2 + \ell^2 w(s)^4 \right) + \varepsilon_3 \ell^2 w(s)^2 \right) ds,
$$

where $u(s)$ and $w(s) = v(s)\theta'(s)$ are the controls, under the dynamical constraints

$$
v'(s) = \frac{1}{v(s)} \left( -\frac{v(s)}{\tau} + f(s) \right), \qquad\qquad v(0) = v^0,
$$

$$
e'(s) = \frac{\sigma(e(s)) - f(s)v(s)}{v(s)} - \frac{\ell(c_{\text{stat}} - \ell w(s))}{v(s)} \left( \frac{v(s)^2}{R_k(s)} \cos \theta(s) - g \sin \theta(s) \right),
$$

$$
e(0) = e^0, \ \ e(s) \geq 0, \ \ e(d) = 0,
$$

$$
f'(s) = \frac{\gamma}{v(s)} \left( u(s)(F_{\max} - f(s)) - f(s) \right), \qquad f(s) \geq 0,
$$

$$
\theta'(s) = \frac{w(s)}{v(s)}
$$

and under the state constraint

$$
f(s)^2 + \left( \frac{v(s)^2}{R_k(s)} \sin \theta(s) + g \cos \theta(s) + \ell w(s)^2 \right)^2 - g^2 \leq f_M^2,
$$

for $s \in [0, d]$. Here, $R_k(s)$ denotes the curvature radius on lane $k$ at distance $s$ from the start. The parameters $\varepsilon_i$ are chosen so that the penalty terms are a small perturbation of the first integral which is the time to cross the finish line. We take $\varepsilon_1 = 0.01$, $\varepsilon_2 = 1$, $\varepsilon_3 = 10$. The problem consists in a control affine system with a cost containing the L2-norm of the control so it is well posed [31].

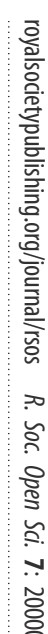

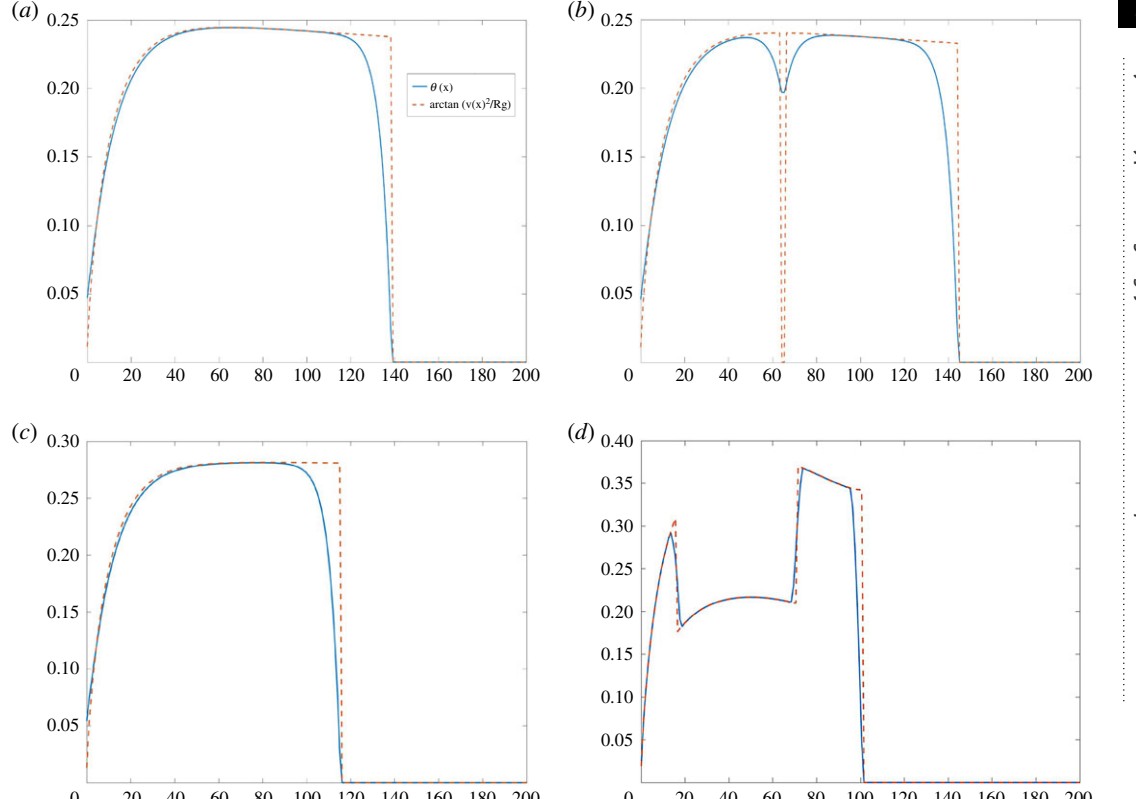

**Figure 6.** The angle $\theta$ (in blue) versus the distance for a runner on lane 5 compared with arctan $(v^2/Rg)$ (in red). (*a*) On a standard track with shorter straights, the maximal angle is 0.25 rad, that is 14°. The runner stays bent longer but at a smaller angle than on the standard track (*c*), for which the maximal angle is 0.28 rad, that is 16.1°. (*b*) On the optimal track, the maximal angle is 0.24 rad, that is 13.7° and adapts to the vertical straight; (*d*) on the double bend track DB2, the angle is much bigger and has to change drastically leading to poor final times.

**Table 1.** Johnson's timesplits for the 1996 World Championships. Line 1, distance in metres; line 2, time splits for 10 m in seconds.

| 20 m | 30 m | 40 m | 50 m | 60 m | 70 m | 80 m | 90 m | 100 m | 110 m | 120 m |
|------|------|------|------|------|------|------|------|-------|-------|-------|
| 3.02 | 0.97 | 0.91 | 0.89 | 0.87 | 0.86 | 0.87 | 0.87 | 0.88 | 0.86 | 0.86 |
|      | 130 m | 140 m | 150 m | 160 m | 170 m | 180 m | 190 m | 200 m | | |
|      | 0.89 | 0.91 | 0.91 | 0.93 | 0.94 | 0.96 | 0.96 | 0.96 | | |

The identification of the parameters is made on Johnson's data given in table 1. This leads to $f_M = 9.5$, $e_0 = 1619.56$, $\tau = 1.2$, $F_{\max} = 16$, $v_0 = 2.32$, $\gamma = 0.0025$.

Optimization and numerical implementation of the OCP are done through the combination of automatic differentiation softwares with the modelling language AMPL [32] and expert optimization routines with the open-source package IPOPT [33]). This allows us to solve for velocity, force, energy, angle in terms of the distance, as illustrated in figure 5 and providing the optimal strategy and the final time.

On a standard track, the runner starts on the semi-circle and he adapts his angle $\theta$ in such a way that $\tan\theta = v^2/Rg$. Then he straightens when reaching the straight part of the track. We have chosen the numerical parameters to match Michael Johnson's record in Atlanta in 1996 to see whether he could have done better on the new optimal track. He was on lane 5 with a final time of 19.32 s. The lane width is 1.22 m and lanes 2 and out are measured 20 cm out from the inside of the lane. For lane 1, according to our model, Johnson would have made 19.37 s on the usual standard track. The optimal track illustrated in figure 3 is indeed significantly more favourable and the time discrepancy between lanes is smaller: he would have beaten his record by four hundredth of a second, with a final time of

19.285 s on lane 5, and 19.30 s on lane 1. The angle of the runner is illustrated in figure 6. On this new track, the runner does not have time to be fully straight between the two curves (difference of angle of four degrees) because the vertical straight is very short.

It is interesting to note that the times we compute on the new track are actually very close to the thousandth to the modified standard track with shorter straights of 61 m that we have drawn in figure 3.

To complete our study, we mention that similar simulations on double bend tracks DB2 lead to a final time of 19.52 s on lane 1 and of 19.484 s on lane 5, while on DB1, it is 19.44 for line 1 and 19.40 for line 5. They are much slower tracks because the outer radius of curvature is very small.

Here, following [30] we have taken $c_{stat} = 0.17$. Assuming that the runner would have more difficulty in staying bent, that is for instance his static coefficient is five times bigger, $c_{stat} = 0.85$, he takes then slightly longer to run and the discrepancy between lanes is bigger but remains only of the order of a hundredth of a second. The difference between lanes remains significantly smaller.

It is an open question to perform a sensitivity analysis in order to measure how the change of parameters could modify performance.

One could also consider applying a bi-level formulation as in [34] instead of minimizing first on the track and then on the runner as we do here. Nevertheless, the aim is to get in the end a track which does not depend on runners.

# 4. Conclusion

In conclusion, it is quite new to be able to compute the optimal geometry of the track and predict the discrepancy in records according to this geometry. Our OCP couples mechanics, energy, neural drive to determine through cost and benefit, the optimal strategy to run a race. Our study highlights that present standard athletic tracks are not the best to break records. Indeed shorter straights and larger radii of curvature could improve the 200 m record possibly by four hundredths of a second. Using the Dubins path problem, the constraint to encompass other sports can be taken into account leading to a new track with shorter horizontal straights and small vertical straights. Our recommendation is to build such tracks in the future.

Data accessibility. This article has no additional data.
Authors' contributions. The authors declare equal contributions.
Competing interests. The authors declare no competing interests.
Funding. The authors declare no funding.
Acknowledgements. The authors are very grateful to Vincent Hakim and Jean-Pierre Nadal for discussions.

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
