## [Reviewer comments · Royal Society Open Science]

Review History

RSOS-200007.R0 (Original submission)

Review form: Reviewer 1

Is the manuscript scientifically sound in its present form?

Yes

Are the interpretations and conclusions justified by the results?

Yes

Is the language acceptable?

Yes

Do you have any ethical concerns with this paper?

No

Have you any concerns about statistical analyses in this paper?

No

Recommendation?

Accept as is

Comments to the Author(s)

The paper is well written and based on solid methodology from optimal control. I especially like the fact that the author consider the discrepancy between the different lanes in the cost. The conclusion is clear. It should be published as is (the simulation section could be expanded a little bit)

Review form: Reviewer 2 (Dante Kalise)**Is the manuscript scientifically sound in its present form?**

Yes

Are the interpretations and conclusions justified by the results?

Yes

Is the language acceptable?

Yes

Do you have any ethical concerns with this paper?

No

Have you any concerns about statistical analyses in this paper?

No

Recommendation?

Accept with minor revision (please list in comments)

Comments to the Author(s)

This paper discusses the design of athletic tracks to enable new records. The authors address this problem in a mathematical optimization framework, resorting to shape optimization and optimal control theory.

In a first step, the shape of the track is optimized based on geometric constraints such as perimeter, the parametrization of straight segments, and the location of a football field at the interior. Depending on the different constraints in play, different optimal designs are presented with varying degree of similarity to existing standards.

After setting an optimal design, the runner performance over the track is analysed as a dynamic optimization problem, accounting for both the physical model of the runner, and their wish to optimize the race. A central point in this approach is the interplay between the track and the dynamics of the runner, which is represented by the modification of these when running over a bend. Model parameters are set to emulate the run of Michael Johnson's 200m record in Atlanta 1996. With these parameters and following an optimal trajectory over the optimal new design, the authors show that a record improvement of up to 4/100 [s] would be possible.

The paper is well-written and easy to follow. It links a very concrete racing problem to classical topics in optimization and optimal control theory such as shape optimization, Dubins paths, and time/energy-optimal control problems. The results are novel, in fact it is shown that reasonably shaped tracks, complying with constraints such as fitting a football field inside, could lead to new records. The latter is backed by testing over real data. I think the quality and novelty of the manuscript merits publication, provided some minor remarks are addressed:

1. Is there anything that can be mentioned concerning the uniqueness of the optimal shapes found in section 2? Are these shapes obtained directly following Dubins paths or do they involve any kind of numerical computations?

2. Are there any other alternative formulations of the max constraint in the third shape problem? Would area + curvature constraints be enough? I think there's something related to the 0.3m rule missing in this constraint.

3. I understand the rationale "track optimization first, runner optimization second", but since the idea is to design a track to break records, an alternative would be to cast a single bi-level optimization problem where the optimal performance over the track enters as a dynamic constraint. In

D. Kalise, K. Kunisch and K. Sturm. Optimal actuator design based on shape calculus, M3AS 28(13)(2018) pp: 2667-2717

we applied a similar approach in a completely different context, leading to a iterative method which would read: fix a track->solve the optimal control problem->compute a sensitivity of the shape (shape derivative) ->update the shape->repeat.

Could you please comment on whether such a formulation could lead to a different design, or is it evident that the one-directional coupling plays no role?

4. What can be said concerning the well-posedness of the optimal control problem in section 3? Are there any uniqueness/optimality guarantees for the solution found?

5. Is there any way to link for final comments in section 3 concerning performance and model parameters with robustness in control? It would seem a relevant issue parameters are expected to vary across the population (although the population of professional athletes must have a reduced variance on the parameters).

6. Minor typos/comments:

- In the first problem description, the 0.3m issue is already present although is explained for the first time by the end of the second problem. Please re-arrange.

- There's a repetition of "the well-known Dubins problem [16,19]" from one paragraph to another in the first problem.

- Shouldn't the constraint in the third problem include the 0.3m somewhere?

- I suggest that for at least one of the three problems in section 2, the formulation is presented as a typical optimization problem min XX s.t. YY

Decision letter (RSOS-200007.R0)

02-Mar-2020

Dear Professor Aftalion

On behalf of the Editors, I am pleased to inform you that your Manuscript RSOS-200007 entitled "How to build a new athletic track to break records" has been accepted for publication in Royal Society Open Science subject to minor revision in accordance with the referee suggestions. Please find the referees' comments at the end of this email.

The reviewers and handling editors have recommended publication, but also suggest some minor revisions to your manuscript. Therefore, I invite you to respond to the comments and revise your manuscript.

- Ethics statement

- Data accessibility

<http://datadryad.org/submit?journalID=RSOS&manu=RSOS-200007>

- Competing interests

- Authors' contributions

- Acknowledgements

- Funding statement

Because the schedule for publication is very tight, it is a condition of publication that you submit the revised version of your manuscript before 11-Mar-2020. Please note that the revision deadline will expire at 00.00am on this date. If you do not think you will be able to meet this date please let me know immediately.

If your manuscript is newly submitted and subsequently accepted for publication, you will be asked to pay the article processing charge, unless you request a waiver and this is approved by

Royal Society Publishing. You can find out more about the charges at <https://royalsocietypublishing.org/rsos/charges>. Should you have any queries, please contact openscience@royalsociety.org.

Kind regards,

Anita Kristiansen
Editorial Coordinator

on behalf of Dr Jose Carrillo (Associate Editor) and Mark Chaplain (Subject Editor)
openscience@royalsociety.org

Reviewer comments to Author:

Reviewer: 1

Comments to the Author(s)

The paper is well written and based on solid methodology from optimal control. I especially like the fact that the author consider the discrepancy between the different lanes in the cost. The conclusion is clear. It should be published as is (the simulation section could be expanded a little bit)

Reviewer: 2

Comments to the Author(s)

This paper discusses the design of athletic tracks to enable new records. The authors address this problem in a mathematical optimization framework, resorting to shape optimization and optimal control theory.

In a first step, the shape of the track is optimized based on geometric constraints such as perimeter, the parametrization of straight segments, and the location of a football field at the interior. Depending on the different constraints in play, different optimal designs are presented with varying degree of similarity to existing standards.

After setting an optimal design, the runner performance over the track is analysed as a dynamic optimization problem, accounting for both the physical model of the runner, and their wish to optimize the race. A central point in this approach is the interplay between the track and the dynamics of the runner, which is represented by the modification of these when running over a bend. Model parameters are set to emulate the run of Michael Johnson's 200m record in Atlanta 1996. With these parameters and following an optimal trajectory over the optimal new design, the authors show that a record improvement of up to 4/100 [s] would be possible.

The paper is well-written and easy to follow. It links a very concrete racing problem to classical topics in optimization and optimal control theory such as shape optimization, Dubins paths, and time/energy-optimal control problems. The results are novel, in fact it is shown that reasonably shaped tracks, complying with constraints such as fitting a football field inside, could lead to new records. The latter is backed by testing over real data. I think the quality and novelty of the manuscript merits publication, provided some minor remarks are addressed:

1. Is there anything that can be mentioned concerning the uniqueness of the optimal shapes

found in section 2? Are these shapes obtained directly following Dubins paths or do they involve any kind of numerical computations?

2. Are there any other alternative formulations of the max constraint in the third shape problem? Would area + curvature constraints be enough? I think there's something related to the 0.3m rule missing in this constraint.

3. I understand the rationale "track optimization first, runner optimization second", but since the idea is to design a track to break records, an alternative would be to cast a single bi-level optimization problem where the optimal performance over the track enters as a dynamic constraint. In

D. Kalise, K. Kunisch and K. Sturm. Optimal actuator design based on shape calculus, *M3AS* 28(13)(2018) pp: 2667-2717

we applied a similar approach in a completely different context, leading to a iterative method which would read: fix a track->solve the optimal control problem->compute a sensitivity of the shape (shape derivative) ->update the shape->repeat.

Could you please comment on whether such a formulation could lead to a different design, or is it evident that the one-directional coupling plays no role?

4. What can be said concerning the well-posedness of the optimal control problem in section 3? Are there any uniqueness/optimality guarantees for the solution found?

5. Is there any way to link for final comments in section 3 concerning performance and model parameters with robustness in control? It would seem a relevant issue parameters are expected to vary across the population (although the population of professional athletes must have a reduced variance on the parameters).

6. Minor typos/comments:

- In the first problem description, the 0.3m issue is already present although is explained for the first time by the end of the second problem. Please re-arrange.
- There's a repetition of "the well-known Dubins problem [16,19]" from one paragraph to another in the first problem.
- Shouldn't the constraint in the third problem include the 0.3m somewhere?
- I suggest that for at least one of the three problems in section 2, the formulation is presented as a typical optimization problem min XX s.t. YY

Author's Response to Decision Letter for (RSOS-200007.R0)

See Appendix A.

Decision letter (RSOS-200007.R1)

06-Mar-2020

Dear Professor Aftalion,

It is a pleasure to accept your manuscript entitled "How to build a new athletic track to break records" in its current form for publication in Royal Society Open Science. The comments of the reviewer(s) who reviewed your manuscript are included at the foot of this letter.

Please ensure that you send to the editorial office an editable version of your accepted

manuscript, and individual files for each figure and table included in your manuscript. You can send these in a zip folder if more convenient. Failure to provide these files may delay the processing of your proof. You may disregard this request if you have already provided these files to the editorial office.

on behalf of Dr Jose Carrillo (Associate Editor) and Mark Chaplain (Subject Editor)
openscience@royalsociety.org

Appendix A

We would like to warmly thank the referees for their careful reading of the manuscript and their comments which have helped to improve and clarify some points.

Referee 1:

We have added a figure with more simulations and explanations.

Referee 2:

1. For Problems 1 and 2: the solution is unique, as an application of Dubins's results and we have stated it more clearly. As for Problem 3, it relies on a numerical computation and we have explained it better, also in relation to Point 2.

2. It would indeed be nice to have another (equivalent) formulation involving area and curvature constraints but we do not know whether this is possible or not. We understand the referee's viewpoint: having a more "easily computable" criterion would be desirable.

We have added a sentence concerning the missing "0.3" and an explanation about the numerical result.

3. As pointed out by the referee, we have decided to do track optimization first and runner second for two reasons:

- firstly, because otherwise the track could become parameter-dependent: in particular it would depend on the runner and on the distance to run (although, of course, one could think of averaging procedures);

- secondly, because in general, it is observed (and commonly admitted) that changing the radius of curvature is not good. For instance in [11], the performance is analyzed on a clothoid track joining the circle to the straight line, and it not worse for the final time, because the radius of the circular part is smaller than a half circle.

We have added a reference to the paper mentioned by the referee, as a relevant and interesting alternative possibility for addressing the problem.

4. We have added explanations on the well-posedness of the optimal control problem in Section 3. We do not know about uniqueness and optimality, because there is no evident convexity property there.

5. We have added a sentence to link parameters to robustness, at the very end of the paper. This provides an interesting issue for further studies.

6. Minor comments have been taken into account.